# Chemometric-Assisted Litmus Test: One Single Sensing Platform Adapted from 1–13 to Narrow pH Ranges

**DOI:** 10.3390/s23031696

**Published:** 2023-02-03

**Authors:** Lisa Rita Magnaghi, Giancarla Alberti, Camilla Zanoni, Marta Guembe-Garcia, Paolo Quadrelli, Raffaela Biesuz

**Affiliations:** 1Department of Chemistry, University of Pavia, Via Taramelli 12, 27100 Pavia, Italy; 2Unità di Ricerca di Pavia, INSTM, Via G. Giusti 9, 50121 Firenze, Italy; 3Departamento de Química, Facultad de Ciencias, Universidad de Burgos, Plaza de Misael Bañuelos s/n, 09001 Burgos, Spain

**Keywords:** pH sensors, chemometrics, PCA, PLS, analytical chemistry, polymer sensors, colorimetric sensors

## Abstract

A novel 3 × 4 colorimetric sensing platform, named the chemometric-assisted litmus test (CLT), has been developed by covalently anchoring commercial pH indicators to ethylene vinyl alcohol (EVOH). The proposed device can be exploited for pH determinations in a wide range from 1 to 13 and in specific narrow ranges, achieving sufficient accuracy and errors below 0.5 pH units. The experimental procedure is simple, quick and reliable; equilibration is reached in less than 2 h, CLT pictures are acquired by a camera, and data treatment is performed applying chemometric techniques such as principal component analysis (PCA) and partial least square regression (PLS) to RGB indices.

## 1. Introduction

The term “litmus” owes its origin to the old Norse word for “dye” and commonly refers to a natural pH-sensitive and water-soluble dye obtained from lichens; originally, litmus was obtained from any lichen species in the Netherlands while, today, it is mainly prepared from the species *Roccella montagnei* from Mozambique and *Dedographa leucophoea* from California [1]. From a chemical point of view, litmus may contain 10 to 15 different dyes and presents a red coloration under acidic conditions (pH below 4.5), while it turns blue under alkaline conditions (pH above 8.3) [1]. This compound has been used as a dye since at least the time of the Vikings, but in the early 14th century, the Spanish scholar, Arnaldus de Villa Nova (1235–1311), was the first person known to use it as a test of acidity, treating paper with this dye and obtaining the oldest form of a pH indicator strip [1,2].

From that time onward, a wide variety of pH papers have been proposed and mass-produced to test the acidity of aqueous solutions, and this cheap, simple, and quick method has become a must-have in laboratory equipment, especially for those applications in which pH measurements by glass electrode are unsatisfactory [1,3,4,5]. Another interesting application for pH papers can be found in the detection of acidic or alkaline volatile compounds in aerosol, but the details of this application are not discussed in this paper [6,7,8,9].

Despite the countless types of pH papers commercially available, these tools can provide only a primary estimation of pH states and ensure neither reversibility nor accuracy in pH measurements; these limitations have paved the way for the development of various optical pH sensors, inspired by traditional pH papers but aimed at overcoming their drawbacks [1,3,4,5].

Moving to the main features of these sensors and limiting the investigation to the most recent papers, different pH-sensitive molecules have been proposed, among which the most common are commercial pH indicators [1,4,5,10,11,12,13,14,15,16], novel pH-sensitive compounds [1,4,5,17,18], natural dyes [1,4,19,20,21,22,23], and luminescent probes [1,3,4,24,25,26,27]. On the other hand, unlike pH papers, these sensors can exploit different solid substrates and, consequently, anchor mechanisms such as physical entrapment, [1,3,4,5,11,19,20,23] sol–gel synthesis [1,4,10,12,13,14,15,16,24] or covalent immobilization [1,4,5], hence offering a fully tunable device depending on the final application. Moving forward, the target pH range represents a crucial aspect that must be considered when developing a novel optical pH sensor: while a plethora of devices for physiological pH range has been proposed, sensors for extreme pH values or sensors covering the entire pH range, ideally, are much fewer in number [1,26,27]. In this regard, it must be underlined that sensing devices for extreme pH play a fundamental role in pH sensing because the error associated with pH measurements at extremely acidic and alkaline values by the means of a pH-meter is definitely not negligible. On the other hand, sensors covering the entire pH range ensure intrinsic versatility since they can be exploited to monitor any chemical reaction or determination, regardless of changes in the pH.

Last but not least, as we hinted before, the lack of reversibility and accuracy represents one of the main drawbacks of traditional pH papers; regarding the first feature, unfortunately, reversibility also still remains the Achilles heel for optical pH sensors since only a few of them are actually fully reversible, while most of them ensure reversibility only in typical conditions or they are theoretically reversible but cannot operate reversibly [3]. As for the latter aspect, nowadays, several strategies have been proposed to overcome the simple coloration comparison typical of pH papers, mainly relying on complex data manipulation, linearization, and other univariate analysis on color indices [1,3,4,5,10,11,12,13,14,15,16,20,22,23,24] or seldom exploiting simple and reliable multivariate approaches [1,4,5,19].

Looking back to the history of this kind of device and keeping in mind the still-open issues in this field, we propose our chemometric-assisted litmus test (CLT), a reversible, accurate, and tunable sensing platform composed of 3 × 4 miniaturized colorimetric sensors covering pH measurements in an extensive range from 1 to 13. These miniaturized colorimetric sensors are prepared by covalent functionalization of commercially available pH indicators to the ethylene vinyl alcohol (EVOH) copolymer, a plastic-based material used in food packaging with interesting features for application in the sensor field such as transparency, water insolubility, processability, and film-forming ability by pressing or extrusion [28,29,30,31,32,33,34,35]. Reversibility is achieved thanks to the combination of covalent immobilization and pH-sensing by classical protonation and deprotonation reactions of pH indicators; meanwhile, the multivariate data treatment, relying on RGB extraction from images and their analysis by chemometric tools such as principal component analysis (PCA) and partial least square regression (PLS) [36,37,38,39] allows for an accurate measurement of pH value with no need of any instrumentation apart from a common smartphone or a camera [37,40]. Finally, as far as the pH range is concerned, the proposed sensing platform in its entirety covers an extremely wide range, namely from 1 to 13. However, a rational strategy is presented to adapt the device to narrow pH ranges, lowering the number of sensing units to the informative ones for the specific narrow range and improving the CLT analytical performances.

## 2. Materials and Methods

Phenol red, *o*-cresol red, thymol blue, *m*-cresol purple, methyl orange, Congo red, bromocresol green, chlorophenol red, bromothymol blue, alizarin red S, alizarin yellow R and Clayton yellow were of analytical reagent grade, purchased from Merck. HNO_3_, NaH_2_PO_4_, NaNO_3_, citric acid, Na_2_HPO_4_, NaHCO_3_, NaOH, MES buffer, PIPES buffer, and EPPS buffer were reagent grade were purchased from Merck. SoarnoL™ D2908 (29% ethylene content, Melt Flow Rate 8.0) was provided by Nippon Gohsei Europe GmbH. Thionyl chloride solution 1 M in dichloromethane, sodium hydroxide in pellets, nitric acid, phosphate buffer, dimethylacetamide (DMA), and dichloromethane (DCM) were purchased by Alfa Aesar.

Pictures of the array were taken by a NIKON COOLPIX S6200 portable camera; a portable led light box (23 cm × 23 cm × 23 cm) was used to guarantee the reproducibility of the photos (PULUZ, Photography Light Box, Shenzhen Puluz Technology Ltd.). UV–Vis spectra were recorded by a Jasco V-750 spectrophotometer, and pH measurements were performed by a pH-meter (Mettler Toledo mod. SevenMulti) equipped with a combined glass electrode (InLab Pro, Mettler Toledo S.p.A., Milan, Italy).

### 2.1. Receptors’ Selection for Chemometric-Assisted Litmus Test (CLT)

A panel of commercially available pH indicators belonging to different classes was selected to cover the widest pH range from 1 to 13. All of these molecules have sulfonic or carboxylic groups in common in their structure, which were both found to be suitable for the covalent anchoring procedure described below, and showed bright colors in both protonated and deprotonated forms. In Table 1, the pH indicators and their log*K*_a_ values in solution, as found in the literature, are reported.

### 2.2. EVOH Covalent Functionalization Procedure

Ethylene vinyl alcohol (EVOH) copolymer is exploited as a solid support and, among the various commercial EVOH copolymers, SoarnoL™ D2908 (29% ethylene content, Melt Flow Rate 8.0), provided by Nippon Gohsei Europe GmbH, is preferred since it ensures higher permeability and thus higher sensitivity and reaction rate, as reported elsewhere [35].

EVOH was functionalized by a 2-step patented reaction reported in the literature [34,45] to obtain advanced materials for sensing applications, henceforth named Dye-EVOH@, in which the reactive dyes are covalently bound to polymeric chains. Dye-EVOH@ was obtained in blocks of irregular shapes, pressed under heating (300 mg, 2000 psi, 30 s, 160 °C) to obtain sensing thin films using a dual-heated plate manual press. From Dye-EVOH@ films, miniaturized circular optodes (0.5 cm diameter) were cut out using a hole punch for paper [34].

### 2.3. Sensing Platform Preparation

The sensing platform was prepared by poking the twelve Dye-EVOH@ miniaturized sensors with a needle and fixing them to an inert plastic support through a thread to obtain the 3 × 4 array shown in Figure 1. The arrays were then dipped in 100 mL solutions at different pH ranging from 1 to 13 and were left to equilibrate for 2 h.

### 2.4. Solutions Preparation and pH Measurement

To develop the multivariate calibration model, solutions at different pH were prepared as follows: 1 L stock solution of defined composition and pH were prepared and then divided into four aliquots of 100 mL, in which a sensing platform was placed. In Table 2, these solutions are listed, and hereafter referred to as training samples. For extreme acid and alkaline pH values, dilute strong acids and bases were exploited while, for intermediate pH, standard buffers were used, adding the proper amount of strong acid or base to obtain the desired pH, respectively to NaH_2_PO_4_, citric acid (labeled as AH_3_ in the table). and NaHCO_3_ 0.01 M solution, avoiding agents with one or more volatile species involved in the acid–base equilibria, which may evaporate during the analysis and modify the composition of the sample. Furthermore, ionic strength was buffered at 0.1 M by adding NaNO_3_ in the solution with a lower content of acid–base species (from B to N).

Subsequently, to test the model robustness in the case of different ionic strength and buffering agents, other solutions, labeled from *a* to *i* and henceforth named as the robustness samples, were prepared either using already employed buffering agents but different ionic strength value (I = 1 M for samples *a–c* and I = 0.01 M for samples *d–f*) or choosing different buffering agents (samples *g*–*i* are prepared using Good’s buffers at I = 0.1 M). In Table 3, the solutions used for the robustness samples are listed.

### 2.5. Color Analysis and RGB Acquisition

After equilibration, Dye-EVOH@ sensing platforms were removed from the solution and dried with common adsorbent paper before picture acquisition. Pictures of the sensors were taken by a NIKON COOLPIX S6200 portable camera equipped with a 1/2.3″ (6.16 mm × 4.62 mm, crop factor 5.6) 16 mpx CCD sensor. A portable led lightbox (23 cm × 23 cm × 23 cm) equipped with 20 LEDs (550LM, color temperature 5500 K) was used to guarantee the reproducibility of the photos (PULUZ, Photography Light Box, Shenzhen Puluz Technology Limited, Shenzhen, China). Setting ISO at the lowest possible for the camera (80) and using the lightbox, all images were acquired at a shutter speed of 1/60 s and an aperture of f/3.2. The white balance was kept constant for all images by setting a white reference point inside the lightbox. The images (4608 × 3456 pixels) were acquired as a .jpg file using a neutral photo profile from the camera. GIMP software [46] was used to acquire the RGB triplets from the .jpg files straight from the camera, manually selecting the region of interest (ROI) by exploiting the “Intelligent Scissors” tool.

### 2.6. Chemometric Approach for pH Measurements in a Range from 1 to 13

CLT was used to first test for pH measurements from 1 to 13. by exploiting solutions containing inorganic buffering agents at a constant ionic strength value. The RGB triplets of the Dye-EVOH@ sensors, acquired for the training samples measurements as previously described, were used as the experimental data while the pH value, measured after equilibration using a pH-meter (Mettler Toledo mod. SevenMulti) equipped with a combined glass electrode (InLab Pro, Mettler Toledo S.p.A.—Milan, Italy), was used as a reference value for the following multivariate data treatment. All of the chemometric data treatments were performed using the open-source software CAT [47].

RGB triplets, referred to as training samples, were first submitted to principal component analysis (PCA) to rationalize the color evolution and, subsequently, partial least square regression (PLS) was applied to develop a quantitative predictive model able to calculate the pH value (independent variable) from the colors of the sensors (dependent variables). In both cases, the RGB matrix composed of 36 columns (three RGB indices per 12 Dye-EVOH@ sensors) and 52 rows (13 stock solutions perfour4 replicates) was only centered, being that the RGB indices are intrinsically scaled. In the case of the PLS application, the RGB matrix was first split into two submatrices, the development and validation dataset (ratio 3:1) for preliminary validation of the model. Several strategies were followed to select the validation dataset such as the selection of (a) one replicate for each pH value, (b) entire blocks of replicates at similar pH values, and (c) venetian blinds, but no significant differences have been highlighted, and model validation has been achieved in any case.

Then, RGB triplets referred to as robustness samples, composed of 36 columns (three RGB indices per 12 Dye-EVOH@ sensors) × 36 rows (9 stock solutions per four replicates), were both projected as an unknown dataset in the PCA score plot and the pH value predicted for these samples by the PLS model is compared to the experimental one for the first estimation of model robustness.

### 2.7. Chemometric Strategy for pH Measurements in Specific Ranges

Moving from wide to narrow pH ranges, informative Dye-EVOH@ sensors need to first be highlighted, also, in this case, using a reliable chemometric-assisted method, here briefly described:Definition of the pH range of interest;Selection of the corresponding rows in the RGB matrix referred to the training samples;Submission of the selected data to PCA after centering pretreatment;Identification of the influent Dye-EVOH@ sensors as the ones with the highest loading values.

Once the informative Dye-EVOH@ sensors were selected for the pH range of interest, a reduced version of the complete 3 × 4 sensing platform could be exploited for the specific need, and the acquired data could be analyzed as previously described.

As an example, the selection of informative receptors and the subsequent PLS model development, validation, and robustness evaluation is described in the pH ranges from 1 to 4 (A), from 3 to 10 (B) and from 9 to 13 (C).

### 2.8. Dye Release from Dye-EVOH@ Sensors

For the sake of brevity, dye release was checked only for two out of twelve Dye-EVOH@; UV–Vis spectroscopy was used to register the spectra at alkaline pH to exploit the intense absorption of the deprotonated form of the test dyes, which were dye no. 3 (596 nm) and no. 7 (617 nm). Dye release was tested by dipping around 100 mg of the Dye-EVOH@ sensors in 25 mL NaOH 0.1 M, registering the UV–Vis spectra after 2, 5, 8, and 10 days, and calculating the amount of dye released.

## 3. Results and Discussion

### 3.1. Visual Analysis of the Sensing Platform’s Color Evolution with pH

As described in Section 2.4, images of the Dye-EVOH@ sensing platforms, after equilibration at different pH, were acquired for each replicate and each training solution described in Table 2. Since the large number of Dye-EVOH@ sensors in the platform and of the platforms exploited (i.e., four replicates × 13 solutions) makes the direct naked-eye analysis of the original pictures complicated as they were shot; visual analysis was carried on by relying on the pictogram displayed in Figure 2, built as follows. For each Dye-EVOH@ sensor at each experimental pH, the average R, G, and B values for the four replicates were calculated. Then, using PowerPoint Software, 12 rectangles of constant dimensions, one per each Dye-EVOH@ sensor in the array, labeled according to the numbering in Table 1, were colored using the “gradient fill” and setting at each experimental pH, with the average RGB triplet as “gradient stop”. The final pictogram well summarizes the color evolution of the Dye-EVOH@ sensor array in a pH range from 0 to 14.

At a glance, we can observe that the color changes of the Dye-EVOH@ sensors were equally distributed within the pH range and at acidic pH, sensors from 1-EVOH@ to 6-EVOH@ were the most informative; at neutral pH, sensors from 7-EVOH@ to 10-EVOH@; and at basic pH, sensors from 1-EVOH@ to 4-EVOH@ and from 10-EVOH@ to 12-EVOH@. Furthermore, comparing the pH values of the color turning of Dye-EVOH@ sensors to those of dyes in solution, the log*K*_a_ value increase of around 1 unit was observed, as already hinted in the literature [34].

### 3.2. Chemometric-Assisted pH Measurements from 1 to 13

After visual analysis, PCA was applied on the RGB triplets of the Dye-EVOH@ sensing platform of the training samples to rationalize the color evolution of the sensors: the PCA model was built considering only the first two components, which explained 69.1% of the experimental variance. Analyzing the loading plot displayed in the Appendix A, we observed that the Dye-EVOH@ sensors that changed color at neutral pH presented a high loading value on PC1, while the Dye-EVOH@ sensors that reacted at both acidic and alkaline pH had the highest contribution on PC2. Moving to the score plot in Figure 3a, the samples displayed an arch-shaped distribution in the plot, from the lower right to the lower left at increasing pH values and three main clusters, partially overlapped, could be identified, depending on the informative Dye-EVOH@ sensors:For pH below 4, samples were separated alongside PC2, and their score value on this component was directly related to pH;For pH between 4 and 9m the samples were mainly separated alongside PC1, and their score value on this component was directly related to pH;For pH above 9, the samples were separated alongside both PC1 and PC2, with the PC1 score value increasing and the PC2 score value decreasing at increasing pH

Even from the PCA score plot, the possibility of calculating the pH value by exploiting this sensing platform appears glaring, since the pH values could be clearly distinguished, and a defined trend was observed when changing the pH value.

In Figure 3b, the projection of the robustness samples on the previously developed score plot is shown: robustness samples were generally located in the correct area of the score plot, thus suggesting that the influence of a buffering agent and ionic strength is negligible.

Second, we developed a PLS model that was able to calculate the pH value (independent variable) from the sensors colors’ (dependent variables); as hinted at before, the training sample matrix was first split into two submatrices, the development and validation dataset (ratio 3:1), for preliminary validation of the model. In the Appendix A, the results obtained using one replicate for each pH value as the validation set were reported, while hereafter, only the final model will be discussed, for the sake of brevity.

The 3-component model was judged to be suitable for our purposes since it ensured 97.80% of the explained variance in cross-validation (CV), 0.5431 of the global root mean square error in CV (RMSECV), and as far as the robustness samples are concerned, 0.4791 of RMSE in prediction (RMSEP). Analyzing the experimental vs. fitted plot reported in Figure 4a, we observed that the training and robustness samples were similarly distributed alongside the y = x line, and no significant difference in the fitting error arose when changing the buffering agent or the ionic strength. Regarding the residuals plot (Figure 4b), the difference between the experimental and fitted values was generally below 0.5 units for both the training and robustness samples, and a random distribution of the residuals around 0 was observed.

### 3.3. Chemometric-Assisted Sensor Selection and pH Measurements in Specific Ranges

We already demonstrated that the chemometric-assisted litmus test could adequately measure the pH in an extremely wide range from 1 to 13, showing an average error in prediction lower than 0.5 units, but in some cases, it might be interesting to focus on more specific pH ranges. Obviously, when focusing on narrow pH ranges, the number of Dye-EVOH@ sensors in the sensing platform can be lowered, and only the actual informative ones need to be highlighted. This selection could be made by just looking at the pictogram reported in Figure 2. In this case, using a reliable chemometric-assisted method is recommended or even mandatory. The sensor selection and model development strategy are summarized in Section 2.7, and hereafter, the procedure for three pH ranges, namely, from 1 to 4 (A), from 3 to 10 (B), and from 9 to 13 (C), will be discussed as an example.

First, the pH ranges must be selected: in our case, we defined three intervals (A, B, and C) reported in the first row of Table 4. Second, the corresponding training samples were highlighted: 16 for interval A, 32 for interval B, and 20 for interval C. For each interval, we started considering all the Dye-EVOH@ sensors as candidates (12 sensors, 36 variables), and we applied PCA to select the actual informative ones. In all cases, the PCA models were built considering only the first two principal components; by analyzing the percentages of explained variance reported in Table 4, we noted that the highest percentage on PC1 was associated with the intermediate pH range while the alkaline and, even more, the acidic one, presented a lower percentage of explained variance on PC1. This could be explained considering the number of reactive dyes in each interval, which is higher for the intermediate one and lower for the other two, as demonstrated later on.

From the scores plot reported in the Appendix A, we could observe that the samples equilibrated at different pH values were separated alongside PC1; therefore, only the loading values on that axis were considered for selecting the informative Dye-EVOH@ sensors per each pH interval. Analyzing the loading values on PC1, as reported in Figure 5, the informative ones for each pH range were clearly detectable as the Dye-EVOH@ sensors with the highest bars. To ease the interpretation of the plots, the name of the pH indicator is reported instead of the sensor number listed in Table 1, and a dashed rectangle highlights the informative ones. Going into detail, the first six sensors were selected for pH range A; Dye-EVOH@ sensors from 7-EVOH@ to 11-EVOH@ were chosen for range B; and the first and the last four for pH range C.

Finally, in Figure 6a–c, the plots reporting the experimental vs. fitted values for each model are shown. As we can see, for all the models, there was good agreement between the experimental and fitted values and no significant difference was observed between the training and robustness samples, confirming the models’ robustness toward the type of buffering agent and ionic strength. To further improve the models’ accuracy and reduce the RMSEs, the reproducibility of the Dye-EVOH@ sensors should be improved, mainly in terms of the sensor thickness. Even the residuals, reported in Figure 6d–f, presented a random distribution.

### 3.4. Evaluation of Dye Release from Dye-EVOH@ Sensors

In Figure 7, UV–Vis spectra registered during the release experiments in the case of 3-EVOH@ (Figure 7a) and 7-EVOH@ (Figure 7b) are reported. A similar behavior was observed for these two dyes, suggesting that all the dyes had similar characteristics, and the release kinetic was mainly determined by the polymeric support or the nature of the covalent bond between the dye and the support.

From the UV–Vis spectra, the dye concentration in solution (μM) and dye release, expressed as μmol per Dye-EVOH@ g, were calculated. In Appendix A, the numerical values of both the dye concentration in solution and dye release are reported, while the plot in Figure 8 graphically displays the trend observed for the dye release in 10 days. Considering the estimated value of 0.1 mmol/g of dye content in Dye-EVOH@, as reported in the literature for similar systems [34], the dye release ranged from 0.8% to 1% for 3-EVOH@ and from 0.4% to 0.8% for 7-EVOH@ of the total dye content in 10 days. Such a low dye release in extremely alkaline conditions is definitely acceptable for the type of device and possible applications; furthermore, the low amount of dye released, the slow release kinetic, and the equilibrium reached in the final days suggest that only dye molecules not covalently bound to EVOH were released in these experiments. This issue can be solved by improving the washing step during the synthetic procedure.

Bearing in mind that the main reason for the lack of reversibility in optical pH sensors is represented by dye leaching since most pH sensing mechanisms are intrinsically reversible, the results we obtained allow us to affirm the complete reversibility of our device thanks to the intrinsic reversible pH indicators chosen and the successful covalent binding between the dyes and the polymetric chains, which prevents extended dye leaching during device utilization.

## 4. Conclusions

A colorimetric sensing platform for pH measurements was proposed, named the chemometric-assisted litmus test (CLT): this device guarantees reversibility, thanks to the covalent binding between the polymeric matrix and receptors and the pH sensing mechanism, wide pH range of application from 1 to 13, and accuracy in pH determination, which is performed by relying on quantitative and predictive chemometric tools, rather than on naked-eye comparison or complex data manipulation.

The design and preparation of the CLT have been described in detail, and both the experimental procedure and the data analysis for pH measurements in a range from 1 to 13 are described; furthermore, a reliable chemometric-assisted strategy for pH measurements in specific ranges is described, and some examples are reported. The dye release in alkaline solutions was investigated, and an equilibrium was reached after around five days with 1% of dye released. This result is definitely satisfying, considering that the covalent bond between EVOH and pH indicators is highly susceptible to an alkaline environment.

The proposed device does represent an interesting tool for pH measurements commonly performed in the lab, since it allows for continuous monitoring, satisfactory accuracy with prediction errors around or below 0.5 even at extreme pH, both acidic and alkaline, and all the procedures for data acquisition and treatment can be easily integrated into a software or mobile application. Therefore, this device can be exploited for the in situ monitoring of pH in several cases such as chemical reactions, biological and biochemical experiments, kinetic experiments, aging investigations, or any other application in which pH values have to be monitored.

## Figures and Tables

**Figure 1 sensors-23-01696-f001:**
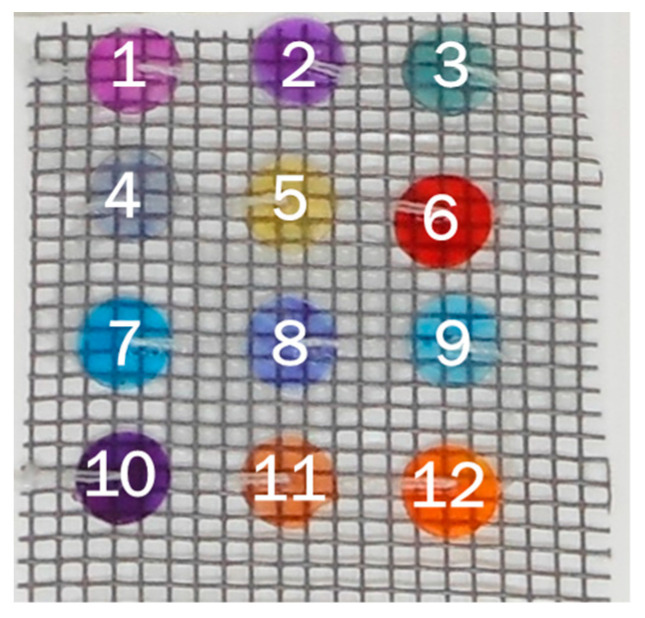
Example of the 3 × 4 sensing platform used as a chemometric-assisted litmus test (CLT). Dye-EVOH@ sensors are numbered from 1 to 12, according to the numbering in Table 1.

**Figure 2 sensors-23-01696-f002:**
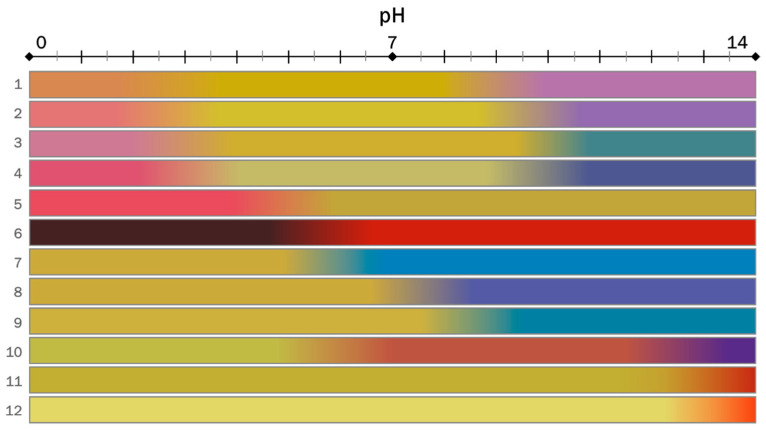
Pictogram reporting the color evolution of Dye-EVOH@ sensors in a pH range from 0 to 14. Colored rectangles are labeled on the left according to the numbering in Table 1, while pH values are reported in the axis above the pictogram.

**Figure 3 sensors-23-01696-f003:**
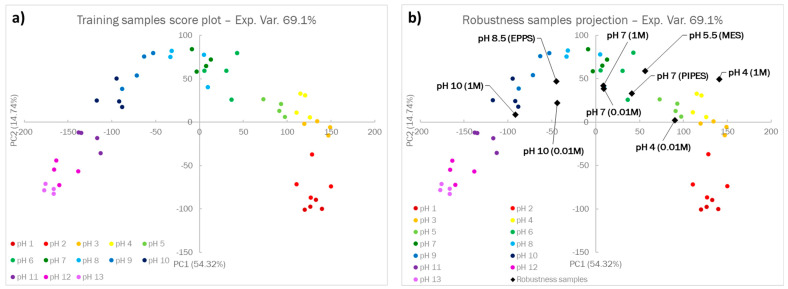
The score plot of the PCA model on the first two principal components, built on the training set (**a**), and the projection of the robustness samples in the previously developed model (**b**).

**Figure 4 sensors-23-01696-f004:**
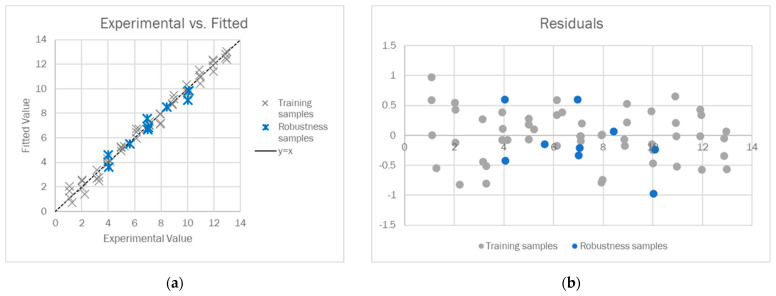
Experimental vs. fitted plot for the training (grey) and robustness samples (blue) (**a**) and the residuals for the training (grey) and robustness samples (blue) (**b**).

**Figure 5 sensors-23-01696-f005:**
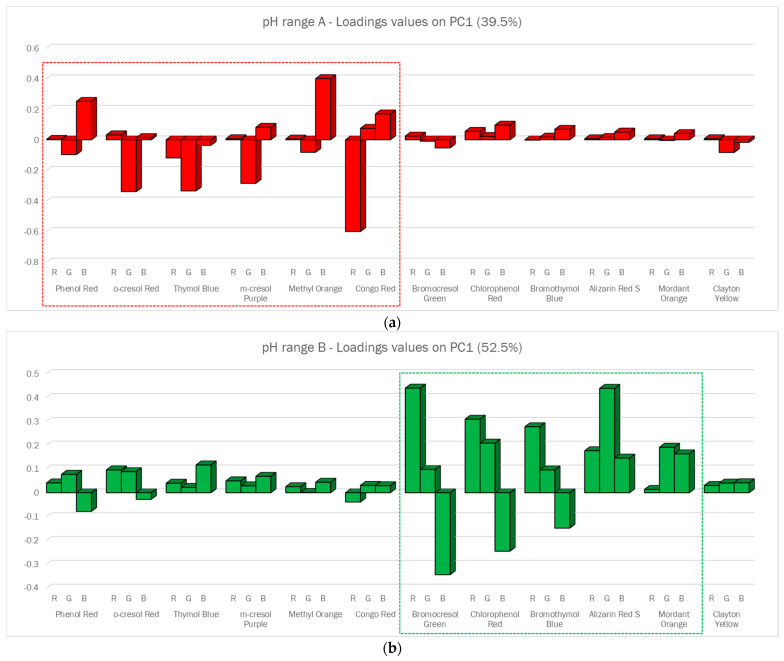
Loading values on PC1 for the PCA models built on the corresponding training samples listed in Table 4 for pH range A (**a**), B (**b**), and C (**c**).

**Figure 6 sensors-23-01696-f006:**
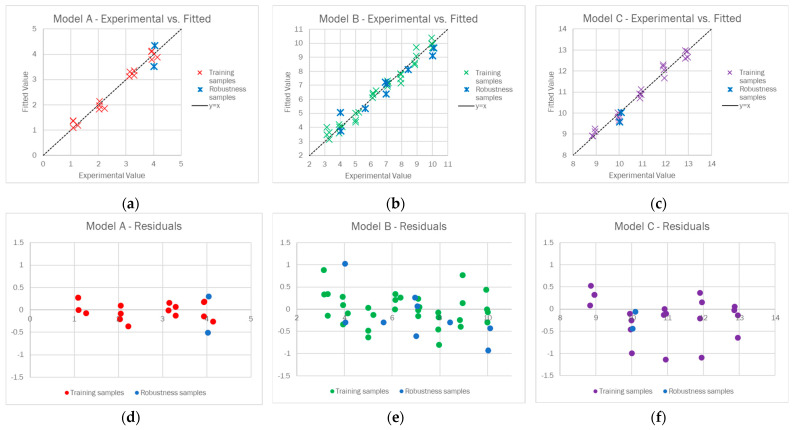
Experimental vs. fitted plot for the training (red for model A, green for model B, violet for model C) and robustness samples (blue) for model A (**a**), model B (**b**), and model C (**c**), and the residuals for training (red for model A, green for model B, violet for model C) and the robustness samples (blue) for model A (**d**), model B (**e**), and model C (**f**).

**Figure 7 sensors-23-01696-f007:**
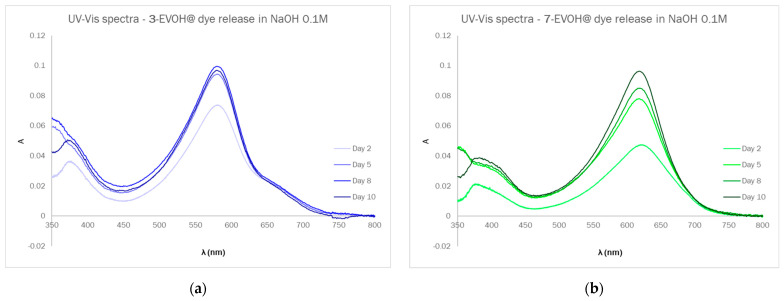
UV–Vis spectra of solutions registered during release experiments in the case of 3-EVOH@ (**a**) and 7-EVOH@ (**b**).

**Figure 8 sensors-23-01696-f008:**
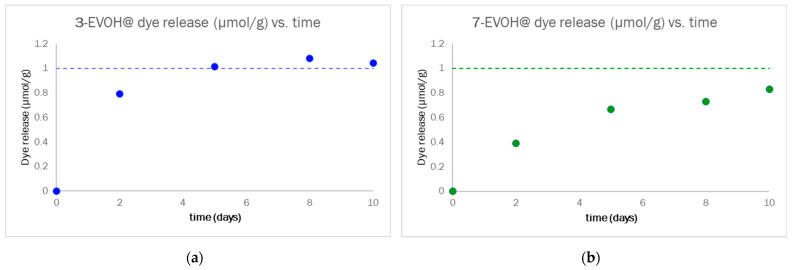
Dye release evolution for 10 days of monitoring for 3-EVOH@ (**a**) and 7-EVOH@ (**b**).

**Table 1 sensors-23-01696-t001:** pH indicators selected for the chemometric-assisted litmus test (CLT), corresponding log*K*_a_ values, and bibliographic references.

n	pH Indicator	log*K*_a_	log*K*_a1_	log*K*_a2_	References
1	phenol red	--	8.32	1.57	[41,42]
2	*o*-cresol red	--	8.20	1.11	[41,42]
3	thymol blue	--	8.9	1.50	[41,42]
4	*m*-cresol purple	--	8.32	1.57	[41,42]
5	methyl orange	3.42	--	--	[43]
6	Congo red	4.1	--	--	[44]
7	bromocresol green	4.35	--	--	[41]
8	chlorophenol red	6.0	--	--	[41]
9	bromothymol blue	7.1	--	--	[41]
10	alizarin red S	--	11.0	4.5	[44]
11	alizarin yellow R	11.5	--	--	[44]
12	Clayton yellow	12	--	--	[44]

**Table 2 sensors-23-01696-t002:** Solutions used for the training samples (A–O).

Solution	Acid-Base Species	Conc. (M)	I Buffer	Conc. (M)	Exp. pH
A	HNO_3_	0.1	--	--	1.12
B	HNO_3_	0.01	NaNO_3_	0.1	2.06
C	H_3_PO_4_-H_2_PO_4_^−^	0.01	NaNO_3_	0.1	3.07
D	Citric acid (AH_2_^−^-AH^2−^)	0.01	NaNO_3_	0.1	3.97
E	Citric acid (AH_2_^−^-AH^2−^)	0.01	NaNO_3_	0.1	4.97
F	Citric acid (AH_2_^−^-AH^2−^)	0.01	NaNO_3_	0.1	6.02
G	H_2_PO_4_^−^-HPO_4_^2−^	0.01	NaNO_3_	0.1	7.01
H	H_2_PO_4_^−^-HPO_4_^2−^	0.01	NaNO_3_	0.1	8.05
I	HCO_3_^−^-CO_3_^2−^	0.01	NaNO_3_	0.1	8.99
L	HCO_3_^−^-CO_3_^2−^	0.01	NaNO_3_	0.1	10
M	HCO_3_^−^-CO_3_^2−^	0.01	NaNO_3_	0.1	11.06
N	NaOH	0.01	NaNO_3_	0.1	12.13
O	NaOH	0.1	--	--	13.04

**Table 3 sensors-23-01696-t003:** Solutions used for the robustness samples (*a*–*i*).

Solution	Acid-Base Species	Conc. (M)	I Buffer	Conc. (M)	Exp. pH
*a*	Citric acid	0.01	NaNO_3_	1	4.04
*b*	Na_2_HPO_4_	0.01	NaNO_3_	1	6.95
*c*	HCO_3_	0.01	NaNO_3_	1	9.82
*d*	Citric acid	0.01	NaNO_3_	0.01	4.08
*e*	Na_2_HPO_4_	0.01	NaNO_3_	0.01	7.08
*f*	HCO_3_	0.01	NaNO_3_	0.01	9.97
*g*	MES	0.01	NaNO_3_	0.1	5.52
*h*	PIPES	0.01	NaNO_3_	0.1	7.00

**Table 4 sensors-23-01696-t004:** pH range, correspondent training samples, candidate Dye-EVOH@ sensors, and % explained variance for pH ranges A, B, and C.

	pH Range A	pH Range B	pH Range C
pH range	1–4	3–10	9–13
Training samples	A–D (16 samples)	C–L (32 samples)	I–O (20 samples)
Candidate Dye-EVOH@ sensors	12 (36 variables)	12 (36 variables)	12 (36 variables)
% Explained variance PC1	39.49%	52.49%	48.24%
% Explained variance PC2	16.81%	10.03%	11.91%

## Data Availability

Not applicable.

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
