# Peer review of "Chemometric-Assisted Litmus Test: One Single Sensing Platform Adapted from 1–13 to Narrow pH Ranges"

_sensors, 2023, doi:10.3390/s23031696_

Round 1

Reviewer 1 Report

Suggestion - A table summarizing/comparing the cost effectiveness of the CLT platform to state of the art pH sensors would be interesting to the readers. I would recommend adding that.

Otherwise, it is a well written paper and I recommend this article for publication.

Reviewer 2 Report

The manuscript by Magnaghi et al. describes a CTL plat form for pH sensing. The topic is interesting and the experiments were well designed. The obtained results show CTL has pretty good performance for sensing pH ranging from 1-13. In general, the manuscript is well organized and the conclusion are supported by the obtained results. In my opinion, it is suitable for publication after addressing the following points.

1. It is necessary to emphasize why pH sesning ranging from 1-13 is important in the introduction part.

2. How about the stability of CTL if the sample desirabe for measurement are varied greatly. Is there any pretreatment needed for complex samples?

3. It is not clear why the samples listed in Table 2 arselected as training samples. Is this special for any specific measurement purposes?

4. Could CTL be directly applied for real complex samples with certain numbers of interferences? For example, the blood serum or the tear.

Reviewer 3 Report

Magnaghi et. al. has reported Chemometric-assisted Litmus Test: one single sensing platform adapted from 1-13 to narrow pH ranges. Authors have mentioned novel 3x4 colorimetric sensing platform, named Chemometric-assisted Litmus Test (CLT), is developed by covalently anchoring commercial pH indicators to ethylene vinyl alcohol (EVOH). This is an interesting article addressing an important issue and very well presented, and is within the scope of this journal. I am sure that this review will receive great attention from the people working in this area and at the same time would serve as a good reference material. I recommend the publication of this article after some minor corrections.

1.     The introduction still does not properly highlight the purpose of the work. The authors have added some irrelevant information about the Chemometric Sensors with application in the introduction.

2.     The language of the manuscript needs to improve, there are some typos and grammatical errors in whole of the manuscript.

3.     Conclusion is not appropriate. It should summarize major findings.

4.     Authors should recheck the abbreviation of the references

Reviewer 4 Report

1. Specific application of the demonstrated pH sensing platform in realistic field should be highlighted in the discussion.

2. Resolution of the Fig. 5 graphs especially labels should be improved.

3. Table 4,  PC1, Loading values in pH range A, B and C has significant variations please explain it. 
